# Low-Rank Compression of Language Models via Gradient-based Rank Selection

## Abstract

Approaches for large-language model compression using low-rank decomposition have made strides, particularly with the introduction of activation and loss-aware Singular Value Decomposition (SVD) that improve the trade-off between decomposition rank and downstream task performance. Despite these advancements, a persistent challenge remains—selecting the optimal ranks for each layer to jointly optimize compression rate and downstream task accuracy. Current methods either rely on heuristics that can yield sub-optimal results due to their limited discrete search space or are gradient-based but are not as performant as heuristic approaches without post-compression fine-tuning. To address these issues, we propose Learning to Low-Rank Compress (LLRC), a gradient-based approach which directly learns the weights of masks that select singular values in a fine-tuning-free setting. Using a calibration dataset of just 3,000 documents, this training architecture teaches the model to select fewer and fewer singular values while minimizing the divergence of intermediate activations from the original model. Our approach outperforms competing fine-tuning-free rank selection approaches, such as Sensitivity-based Truncation Rank Searching (STRS), Adaptive Rank Selection (ARS), and LLM-Pruner on Llama-2-7B, Llama-3-8B, Gemma-7B, and Llama-2-13B across various compression rates on common-sense reasoning and open-domain question-answering tasks; For instance, with a compression rate of 20%, our approach outperforms the competitive STRS on MMLU, BoolQ, and OpenbookQA by 12%, 3.5%, and 4.4%, respectively, using Llama-2-13B. More remarkably, our fine-tuning-free approach consistently outperforms LLM-Pruner, even after fine-tuning, on NQ-Open, MMLU, BoolQ and OpenbookQA with Llama-2-7B.

## 1 Introduction

Large language models (LLMs) such as GPT-3 (Brown et al., 2020) and LLaMA (Touvron et al., 2023) are showing remarkable results in natural language understanding and generation tasks. These models are not just pivotal in zero-shot language modelling but also extend their utility to applications such as code generation (Chen et al., 2021), conversational agents (Kumar et al., 2023), and personalised education (Kasneci et al., 2023). Despite the success of these models in solving a wide range of tasks, their use is limited by high computing and memory requirements. For example, LLaMA-2 comes in 7 billion and 40 billion parameter variants (Touvron et al., 2023), requiring 25.79 GB and 153.87 GB of memory, respectively. Yet, these models are small compared to others, such as GPT-3 (Brown et al., 2020) and PaLM (Chowdhery et al., 2022), which have 175 billion and 500 billion parameters, respectively.

As these models grow, various compression techniques have been developed to reduce their size. Quantization, which reduces the number of bits required to represent each parameter, is widely used for compressing language models (Dettmers et al., 2022; Frantar et al., 2023; Lin et al., 2024). For instance, `LLM.int8()` is a procedure for using 8-bit precision in matrix multiplication in transformer layers that had cut inference memory requirements by half without significant performance degradation (Dettmers et al., 2022).

As an alternative to quantisation, other works explored the structural pruning of LLMs (Ma et al., 2023; Xia et al., 2024). These works follow two stages for compression: first, pruning for model

compression and then a continued training step to recover performance. LLM Pruner (Ma et al., 2023) utilizes parameter-efficient fine-tuning after compression to recover performance whereas Sheared Llama (Xia et al., 2024) performs extensive training on 50 billion tokens after compression. Other recent works show that applying low-rank matrix decomposition techniques can compress and reduce memory requirements of language models (Hsu et al., 2022a; Li et al., 2023; Yuan et al., 2024). Moreover, the application of these methods has seen improvements through weighted singular value decomposition, a variant that makes the decomposition loss aware or activation aware (Hsu et al., 2022a; Yuan et al., 2024), thus improving the trade-off of compression and performance. Yet, one key challenge that both approaches face is the selection of optimal ranks for each layer, which determines the amount of compression. Given that research has shown that different layers of a language model may have different optimal compression rates Yuan et al. (2024); Nawrot et al. (2024), using a constant rank across layers may not be an effective solution. To address this problem, Yuan et al. (2024) propose a heuristic named Sensitivity-based Truncation Rank Searching (STRS), which iteratively searches for optimal ranks per layer by evaluating model perplexity on a small calibration set. Despite proving better than a naive selection of constant compression ratios across layers, this approach has two core problems that can lead to suboptimal solutions. First, the search space of ranks is a discrete set of only 10 elements, significantly restricting the number of options available for the ranks. Second, the optimal decomposition of each layer is identified independently, without taking into account the decomposition of the other layers. On the other hand, Gao et al. (2024)

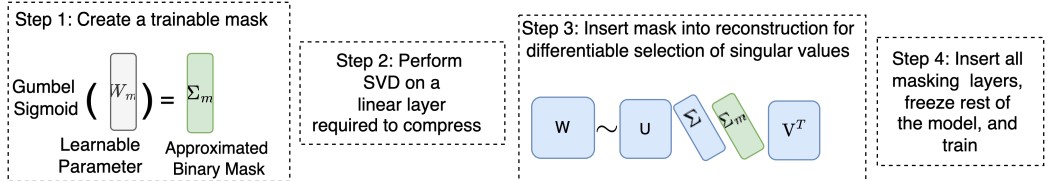

Figure 1: Outline of our proposed method to learn SVD ranks for low-rank compression.

proposed a rank selection approach to learn the optimal ranks through gradient descent. In Adaptive Rank Selection (ARS), a binary masking mechanism is used for optimizing the number of ranks through training Gao et al. (2024). Using GRUs (Dey & Salem, 2017) and linear projections, ARS introduced a learnable singular value masking layer into the SVD reconstruction from which the rank was extracted. However, a key shortcoming is that rank selection using ARS leads to heavy performance degradation and requires an expensive post-compression training stage. Similar to this approach, pruning techniques as Wang et al. (2020c) have also explored learning singular value masking for compression; however, unlike ARS, it performs training of the entire model and focuses on smaller models like BERT. To address these problems in rank selection in low-rank decomposed models, we propose a method called *Learning to Low-Rank Compress* (LLRC).

LLRC can learn the optimal per-layer factorisation ranks by introducing a singular value selection mask $\Sigma_m$ into the matrix reconstruction, which is optimised via gradient-based optimisation; Fig. 1 provides a high-level outline of the method. The mask $\Sigma_m$ is trained using a multi-objective loss function that enables the balancing of compression costs and downstream task performance.

This training approach is lightweight, as it only requires gradient computation for the linear singular value masking layers rather than for the entire model. Following training, after the overall desired compression rate is achieved, the masks are directly utilised to select the most optimal number of singular values for the given compression rate. To summarise, our key contributions are the following:

- A fine-tuning free technique for LLMs called *Learning to Low-Rank Compress (LLRC)* to learn optimal singular values for each layer through training on a small calibration dataset.
- A learnable singular value masking linear parameter which learns, in a fine-tuning free setting, to select the most optimal *any-k* singular values for compression of LLMs.

## 2 RELATED WORK

Model compression is a crucial field in deep learning that focuses on reducing the computational costs associated with deploying models while maintaining their performance. There are several

approaches to neural network compression, including pruning (Han et al., 2015; Zhu & Gupta, 2017; Ma et al., 2023), quantisation (Bai et al., 2024; Xiao et al., 2023), and low-rank factorisation, the focus of this work. Specifically in natural language processing (NLP), there have been various efforts along these lines. Early work aimed to reduce the number of parameters in LSTMs. For instance, Winata et al. (2019) applied low-rank decomposition techniques such as Semi-NMF and SVD for LSTM compression and compared the results to pruning. In contrast to performing low-rank factorisation on a trained model, other works applied tensor decomposition to re-parameterize the model architecture for training (Yang et al., 2017; Pan et al., 2019; Zangrando et al., 2023). For instance, TR-LSTM utilised the low-rank tensor ring decomposition (TRD) to reformulate the input-to-hidden transformation (Pan et al., 2019). While Tensor Train (Yang et al., 2017) and Tensor Ring (Pan et al., 2019) are effective model compression techniques, their application to compress an already trained model is not straightforward.

More recent works focused on applying low-rank factorisation to compress language models without out full re-training. For example, Hsu et al. (2022a) develop a weighted singular value decomposition approach called Fisher-Weighted SVD (FWSVD) that tries to preserve the model performance on a given downstream task. The authors compare approaches using SVD, SVD with fine-tuning, FWSVD, and FWSVD with fine-tuning and show that FWSVD with fine-tuning yields more accurate results than naive SVD with fine-tuning (Hsu et al., 2022a). Activation-Aware Singular Value Decomposition (ASVD) uses the intermediate activations in the weighted singular value decomposition (Yuan et al., 2024), minimising the reconstruction error of the output of the linear transformation rather than minimising the error of the weight itself (Yuan et al., 2024). This method yields more accurate results than Fisher SVD and allows for increased compression at a better trade-off with performance (Yuan et al., 2024).

In both these approaches, while performing low-rank decomposition for compression, the singular value rank for each layer has to be determined. Besides a naive approach that selects an equal rank across all layers, recent works explored approaches to find the optimal rank for each layer. A relevant approach is Sensitivity-based Truncation Rank Searching (STRS; Yuan et al., 2024), which iteratively searches for optimal ranks per layer by evaluating model perplexity on a small calibration set. STRS performs a binary search by using a discrete set of pre-defined 10 compression rates and iteratively applying low-rank decomposition to one layer, leaving the rest of the network unchanged (Yuan et al., 2024). Whereas, ARS Gao et al. (2024) tackle rank selection using gradient-based optimization, employing a recurrent neural network and linear layers to predict binary masks over singular values.

# 3 BACKGROUND

We now describe how Singular Value Decomposition (SVD) can be used to compress linear layers. Moreover, we also briefly cover details of ASVD (Yuan et al., 2024), a weighted SVD technique which yields higher model compression rates at lower performance loss.

## 3.1 COMPRESSING TRANSFORMERS WITH SVD

SVD is a fundamental matrix factorization technique used in various fields, including image processing and machine learning (Chen, 2018; Yang, 2021; Schotthöfer et al., 2022). The SVD decomposes a matrix $W$ into the product of three other matrices: $U_r$, $\Sigma_r$, and $V_r^T$ as shown in Eq. (1):

$$W = U_r \Sigma_r V_r^T \tag{1}$$

Here, $W \in \mathbb{R}^{m \times n}$ is a rank-$r$ matrix, $U_r \in \mathbb{R}^{m \times r}$ and $V_r \in \mathbb{R}^{n \times r}$ are orthogonal, $\Sigma_r \in \mathbb{R}^{r \times r}$ is the diagonal matrix of the singular values, and $r$ is the rank of the decomposition. Decreasing $r$, thus discarding some of the singular values, reduces the dimension of $U_r$ and $V_r$ but worsens the approximation of $W$. In particular, for any $k \leq r$, it holds:

$$U_k \Sigma_k V_k^T = \arg \min_{\text{rank}(W_k)=k} \|W_k - W\|_2 \,.$$

If we approximate $W$ using a low-rank SVD, then we only need to store the highest $r$ singular values and their corresponding singular vectors. This would result in the storage of three smaller matrices,

$U$, $\Sigma$, and $V$, in place of a larger matrix $W$. Therefore, if $W$ is of shape $(m, n)$, the compression can be quantified by the parameter ratio in Eq. (2), where $r$ denotes the rank of decomposition:

$$\text{Param Ratio} = \frac{m \cdot r + n \cdot r}{m \cdot n}. \tag{2}$$

Such approximation can be applied to linear layers in a transformer, such as key, value, and query projection matrices, and the linear layers in a feed-forward network.

## 3.2 ACTIVATION-AWARE SVD

Activation-Aware SVD is a form of weighted SVD that aims to minimise the reconstruction error of the output of a linear transformation rather than minimising the error of the reconstructed weight matrix (Yuan et al., 2024). This approach can be formulated as optimising the following quantity:

$$\arg \min_{\text{rank}(W_k)=k} \|W_k X - W X\|_2 \tag{3}$$

where $W_k$ is the reconstructed weight using $k$ singular values and $X$ is an input into the linear transformation. This is achieved by performing SVD on $WS$ rather than $W$ and then scaling the reconstructed weight by $S^{-1}$, where $S$ is a matrix calculated from a set of inputs $X$ designed to capture the influence of the input channels on the weights (Yuan et al., 2024).

## 4 LEARNING TO LOW-RANK COMPRESS

In this work, we propose Learning to Low-Rank Compress (LLRC), an approach that learns optimal Singular Value Decomposition (SVD) ranks for each layer from data to achieve a desired compression rate. The approach centres on applying SVD to each weight matrix targeted for compression, followed by a learnable masking layer in the weight reconstruction that selectively masks singular values, as shown in Fig. 1. The training process begins with an uncompressed model, and through training, the model increases the compression until the desired overall compression rate is reached. During training, these masking layers can adaptively learn highly different compression rates for each layer.

The advantage of LLRC is that it jointly optimises compression and model performance, where the model is trained by optimising a multi-task training objective composed of compression, distillation, and mask smoothing loss. During training, the only learnable parameters are the weights that control the masking while the rest of the model is frozen. The first part of the objective is a compression loss that minimizes the average value of the weights responsible for masking, generating more sparsity in the mask. The second part of the objective is a distillation objective, which minimizes the divergence between two intermediate activations between the compressed model and the uncompressed model. The third part of the objective is a Total Variation loss that is applied to the learnable masking layer to guide the learning of smooth masks. The following sub-sections outline further provide details of the training procedure.

## 4.1 APPLYING SVD

We perform SVD on all linear projection layers in the model except for the logits layer. Given the ability of weighted SVD approaches to retain performance at higher compression rates (Hsu et al., 2022b; Yuan et al., 2024), we utilize ASVD as a drop-in replacement for SVD in most of our experiments. For consistency with Yuan et al. (2024), we use $\alpha = 0.5$.

## 4.2 TRAINABLE SINGULAR VALUE SELECTION

Following decomposing a linear layer into its factors using SVD as in Eq. (1), we introduce a learnable mask inserted in its reconstruction, outlined in Fig. 1 and defined as follows:

$$W = U\Sigma(\Sigma_{\text{mask}})V^T \quad \text{with} \quad \Sigma_{\text{mask}} = g(W_{\text{learnable}}). \tag{4}$$

Here, the matrix $W_{\text{learnable}} \in \mathbb{R}^{1 \times \text{rank}}$, used to generate $\Sigma_{\text{mask}}$, is a learnable parameter. The function $g$ in Eq. (4) represents the Gumbel-Sigmoid function (Jang et al., 2017). The Gumbel-Sigmoid

function is a continuous relaxation of discrete Bernoulli random variables, enabling gradient-based optimization over the binary mask in Eq. (4) that leverages the Gumbel-Softmax trick to reparameterise discrete sampling into a differentiable function of the logits (Jang et al., 2017). The learnt mask is represented by $\Sigma_{\text{mask}} \in \{0, 1\}^{1 \times \text{rank}}$ and the reconstructed weight is $W \in \mathbb{R}^{m \times n}$.

At the start of training, $W_{\text{learnable}}$ is initialized such that the mask selects all singular values and through training, its parameters learn sparser masks to achieve the target compression rate. During training, the only learnable parameters in the model are the newly introduced masking layers and the rest of the model is frozen, keeping the training process efficient.

### 4.3 TRAINING

#### 4.3.1 DISTILLATION DATASET

On the training corpus, a set of 3000 documents, we create a distillation dataset which contains the hidden state of each token from the middle layer and the hidden states from before the logits layer (Wang et al., 2020a). The activations in this dataset will serve as labels used during training.

#### 4.3.2 OPTIMISATION

The objective function focuses on minimising the number of singular values selected, minimising the divergence between the activations of the original and compressed models, and enforcing the smoothness of the learned masks. We aim to minimise the total loss $L$, which is a weighted sum of the compression loss $L_{\text{compression}}$, the distillation loss $L_{\text{distillation}}$, and the Total Variation loss $\mathcal{L}_{\text{tv}}$:

$$\mathcal{L} = \alpha \mathcal{L}_{\text{distillation}} + \beta \mathcal{L}_{\text{compression}} + \gamma \mathcal{L}_{\text{tv}}, \tag{5}$$

where $\alpha, \beta, \gamma \in \mathbb{R}$ are user-defined hyperparameters, further discussed in Section 5.4.

**Compression Loss** For the compression loss, we directly minimise the mean of the learnable weights, helping generate sparser masks and increasing the compression rate.

$$\mathcal{L}_{\text{compression}} = \frac{1}{N_{\text{layers}}} \sum_{i=1}^{N_{\text{layers}}} \text{Average}(W_{\text{learnable},i}) \tag{6}$$

Since we minimise only the overall compression rate rather than setting a target compression for each layer, the model can learn different compression rates for each layer type and layer number.

**Distillation Loss** The distillation loss minimises the differences in activations between the original model and the model being compressed using the mean-squared error loss in Eq. (7), following recent model compression results achieved through deep self-attention distillation (Wang et al., 2020b;a). We use the following distillation loss:

$$\mathcal{L}_{\text{distillation}} = \|A_{\text{compressed}} - A\|_F^2, \tag{7}$$

where $A_{\text{compressed}} \in \mathbb{R}^{L \times D}$ and $A \in \mathbb{R}^{L \times D}$ denote the activations of the compressed and original models, respectively, where $L$ is the sequence length and $D$ is the dimension of the hidden state. Following Wang et al. (2020a), we use the activations of the middle layer and the hidden states before the logits layer as targets. As an alternative to distillation, we also conducted experiments using a pre-training next token prediction loss, which can be found in Ablations Section 7.3.

**Total Variation Loss** As described in Eq. (8), we introduce a Total Variation (TV) loss on the learnable mask from Eq. (4) to guide learning a smooth mask. Here, $n$ refers to the index of the mask vector of length $N$. A smooth mask refers to one that selects neighbouring singular values together rather than skipping intermediate values.

$$\mathcal{L}_{\text{tv}} = \sum_{n=0}^{N-1} |\Sigma_{\text{mask},n+1} - \Sigma_{\text{mask},n}| \tag{8}$$

Theoretically, reconstructing a weight matrix after SVD involves utilizing the top-k largest singular values sorted by their magnitude. Following this intuition, if, for example, the 5th and 7th singular

values are deemed relevant by the mask, this loss forces the 6th singular value also to be deemed relevant by the mask. We experimentally analyse the contribution of the TV loss in our ablations in Section 7.2.

## 4.4 MODEL POST-PROCESSING

After training, we use the learned singular value masks to select the required singular values for compression. During the evaluation, we applied a 0.5 threshold to the sigmoid of the logits to generate the binary mask without needing the differentiable masking operator.

If a layer is not compressed more than 1%, we reconstruct its entire weight matrix using its full rank, and the layer remains uncompressed. The early stopping criteria while training accounts for this and only terminates training when the precise compression rate is achieved. This design choice of avoiding using the mask is crucial; even if the learned singular value mask retains 90% of the singular values, compression might not be achieved as per Eq. (2), and the full rank can be used to reconstruct the weight.

## 5 EXPERIMENTAL SETUP

### 5.1 DATASET AND EVALUATION

We used a calibration dataset of 3000 documents from WikiText-2 (Merity et al., 2016) for training. We evaluate the compressed model on zero-shot PIQA (Bisk et al., 2020), BoolQ (Clark et al., 2019), and OpenbookQA (Mihaylov et al., 2018), MMLU (Hendrycks et al., 2021) and 5-shot evaluations on NQ-Open (Kwiatkowski et al., 2019). Datasets cover common-sense reasoning datasets and open-domain QA.

### 5.2 TRAINING AND DATA CONFIGURATION

The training dataset for LLRC consisted of 3,000 unique documents from WikiText-2. To have a greater number of token activations present per batch, all documents selected had more than 150 words. Experiments were conducted using a batch size of 4, a maximum token length of 256, and optimized with the AdamW optimizer (Loshchilov & Hutter, 2019). We employed an initial learning rate of 0.01, which was halved upon reaching the target parameter ratio, to improve training stability. To avoid unnecessary training, we use early stopping, terminating the training 750 steps after the target parameter ratio was achieved.

### 5.3 MASKING LAYER INITIALIZATION

At the start of the training, the learnable weight matrix $W_{\text{learnable}}$ is initialized such that the model starts in an uncompressed state, selecting all singular values. Given that higher singular values are theoretically more significant, we introduce an inductive bias into the weight initialization of the mask. To this end, $W_{\text{learnable}}$ is initialized with linearly distributed values ranging in $[3, 6]$, aligned with the magnitude of the singular values. Following Nawrot et al. (2024), we use a temperature of 0.1 for the Gumbel function, which controls the hardness of the mask.

### 5.4 OBJECTIVE FUNCTION CONFIGURATION

The objective function in Eq. (5) represents a weighted sum of the compression, distillation, and total variation loss. The weight on the compression loss, denoted by $\beta$, is set to a constant value of 1 until the target compression ratio is achieved, after which it is set to 0 to prevent further unnecessary compression and focus on the model loss. Instead of using a constant value for $\alpha$, which is more susceptible to the initial choice, inspired by Fu et al. (2019), we oscillate $\alpha$ between two bounds between 1 and 0 using the cosine function. This also enables the training to oscillate focus between optimizing for compression and performance. More details on the scaling function of $\alpha$ can be found in Appendix A.1.

# 6 RESULTS

The following subsections cover the downstream evaluation performance of our approach. We benchmark it against other models compressed using low-rank decomposition, as well as alternative compression methods such as structural pruning.

## 6.1 EVALUATION BENCHMARKS

To evaluate the efficacy of our gradient-based rank selection training procedure, Learning to Low-Rank Compress (LLRC), we benchmark it against a baseline rank selection and advanced techniques such as Sensitivity-based Truncation Rank Searching (STRS; Yuan et al., 2024) and Adaptive Rank Selection (ARS; Gao et al., 2024). [1] We perform extensive compression performance comparisons on four architectures of different sizes: Llama-2-7B, Llama-3-8B, Gemma-7B, and Llama-2-13B.

Given that weighted SVD approaches have shown better compression results compared to plain SVD (Hsu et al., 2022a; Yuan et al., 2024), in our experiments, we use Activation Aware SVD (Yuan et al., 2024) as a drop-in replacement for SVD to perform low-rank decomposition. As a baseline algorithm for rank selection, we selected ranks equally across all layers that lead to the target parameter ratio of compression—we refer to this approach as "Fixed Rate". If the target model parameter ratio is 0.90, each layer was compressed equally by 10%. To compare fine-tuning free rank selection approaches, for ARS, we only use the rank selection portion of the algorithm without the post-training fine-tuning.

| Method | Param Ratio | LLaMA-2-7b | | | | | LLaMA-3-8b | | | | |
|---|---|---|---|---|---|---|---|---|---|---|---|
| | | NQOpen | MMLU | BoolQ | PIQA | OQA | NQOpen | MMLU | BoolQ | PIQA | OQA |
| Baseline | 1.00 | 26.0 | 41.3 | 77.8 | 78.1 | 31.4 | 29.0 | 62.1 | 81.3 | 79.7 | 34.8 |
| Fixed Rate | 0.90 | 3.41 | 26.3 | 56.3 | 67.7 | 23.2 | 3.10 | 37.6 | 74.6 | 70.1 | 23.2 |
| STRS | 0.90 | 21.2 | 37.6 | 75.8 | 77.5 | 31.8 | 17.7 | 49.7 | 73.2 | 77.7 | 31.4 |
| ARS | 0.90 | 8.01 | 29.0 | 51.9 | 72.2 | 25.4 | 9.09 | 34.8 | 66.6 | 73.9 | 27.6 |
| **Ours** | 0.90 | 22.1 | 37.8 | 77.3 | 78.1 | 33.4 | 19.6 | 44.8 | 71.9 | 77.7 | 32.4 |
| Fixed Rate | 0.85 | 1.94 | 23.7 | 49.9 | 65.2 | 21.2 | 1.69 | 29.0 | 64.6 | 65.2 | 19.6 |
| STRS | 0.85 | 16.3 | 32.4 | 75.1 | 76.0 | 31.4 | 4.76 | 33.0 | 68.6 | 70.2 | 22.8 |
| ARS | 0.85 | 4.40 | 23.1 | 51.6 | 69.9 | 23.0 | 0.17 | 22.9 | 39.3 | 60.8 | 16.5 |
| **Ours** | 0.85 | 18.5 | 33.2 | 74.8 | 77.2 | 32.4 | 14.0 | 29.4 | 61.4 | 75.5 | 29.6 |
| Fixed Rate | 0.80 | 0.53 | 23.7 | 60.0 | 62.9 | 20.8 | 0.33 | 24.9 | 63.0 | 60.9 | 17.4 |
| STRS | 0.80 | 9.25 | 28.1 | 71.6 | 71.2 | 28.2 | 0.25 | 24.9 | 49.1 | 63.3 | 16.6 |
| ARS | 0.80 | 0.89 | 23.0 | 59.1 | 65.3 | 19.8 | 0.17 | 24.0 | 60.0 | 62.9 | 18.2 |
| **Ours** | 0.80 | 13.5 | 29.3 | 71.4 | 75.1 | 32.8 | 8.59 | 24.8 | 53.5 | 74.1 | 25.4 |

Table 1: Comparison of evaluation performance using different rank selection methods on LLaMA-2-7b and LLaMA-3-8b. We compare Fixed Rate (naive baseline), STRS Yuan et al. (2024), ARS Gao et al. (2024), and our proposed approach.

Table 1 shows a comparison of the evaluation performance of these approaches on Llama-2-7B and Llama-3-8B, while Figure 2 contains a visualization of the performance on all the models, evaluated on zero-shot MMLU, BoolQ, PIQA, OpenbookQA and 5-shot NQ-Open. The results in Table 1 show that our proposed approach demonstrates superior performance on 3 out of 5 datasets, NQ-Open, PIQA, and OpenbookQA, compared to all rank selection approaches and performs competitively on MMLU and BoolQ. It consistently maintains higher accuracy across various compression levels on a majority of the datasets.

In particular, our approach exhibits significant advantages at lower parameter ratios. For instance, at a parameter ratio of 0.80 of Llama-2-7B, our approach outperforms STRS on NQopen by 4.3%, on PIQA by 3.9% and on OpenbookQA by 4.6%. For the same parameter ratio, on Llama-2-13B, as shown in Fig. 2, our approach strongly outperforms the competitive STRS on MMLU, BoolQ, and OpenbookQA by 12%, 3.5%, and 4.4%, respectively. In the compression of Gemma-7B, on

---

[1] ARS Implementation: link to code; there is an implementation caveat with ARS, refer to Appendix F.

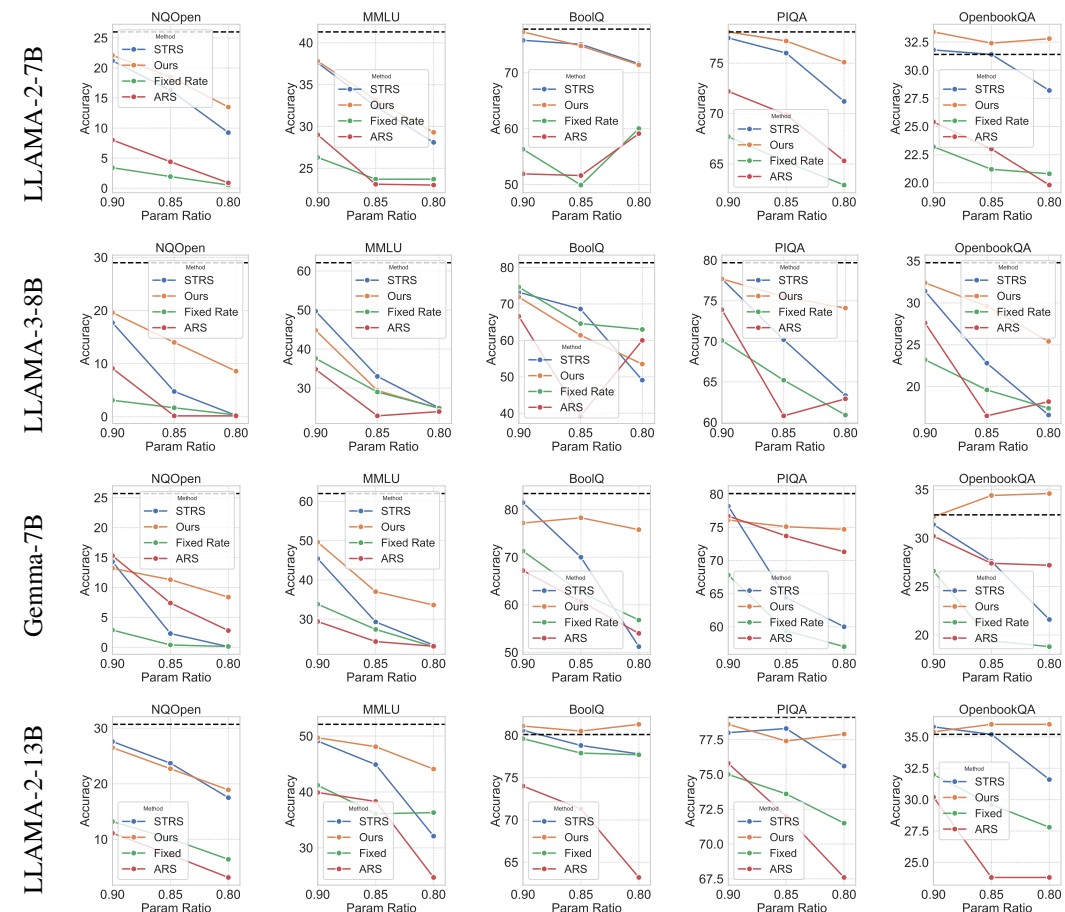

Figure 2: Evaluation of compression performance on four architectures using different rank selection methods

Openbook-QA, our approach leads to an improvement in performance as the parameter ratio decreased from 0.90 and 0.80. We notice that a similar phenomenon was observed by e.g. Sharma et al. (2024), where they showed that selectively reducing the ranks of individual layers can lead to improved 0-shot performance.

Detailed tables of performance metrics for Gemma-7B and Llama-2-13B can be found in Appendix D and Appendix E, respectively. Another pattern we observe is that the performance loss, across all rank selection approaches, is greater in models such as Llama-3-8B and Gemma-7B compared to Llama-2-7B. We hypothesize that this is due to the combination of the feedforward network (FFN) being the least low-rank compressible and the FFN covering a greater percentage of parameters in these models compared to Llama-2-7B. We further explore this in Appendix C.

## 6.2 IMPACT OF FINE-TUNING

To place our work in the larger context of model compression, we benchmark our approach against LLM-Pruner (Ma et al., 2023), a structure pruning method with both fine-tuning and fine-tuning-free variants. Since our approach does not update model weights, we compare it to LLM-Pruner's fine-tuning-free variant.

The comparative analysis presented in Table 2 highlights that our fine-tuning free approach outperforms the fine-tuning free LLM-Pruner results on most datasets across all compression rates. The hyperparameters used to generate the LLM Pruner results are present in Appendix B.3 This demonstrates that our rank selection approach better preserves performance in a fine-tuning-free manner. At a parameter ratio of 0.80, our approach on LLaMA-2-7B not only surpasses LLM-

| Method | Param Ratio | LLaMA-2-7b | | | | | LLaMA-3-8b | | | | |
|---|---|---|---|---|---|---|---|---|---|---|---|
| | | NQ-Open | MMLU | BoolQ | PIQA | OQA | NQ-Open | MMLU | BoolQ | PIQA | OQA |
| Baseline | 1.00 | 26.0 | 41.3 | 77.8 | 78.1 | 31.4 | 29.0 | 62.1 | 81.3 | 79.7 | 34.8 |
| Pruner | 0.90 | 15.5 | 28.5 | 64.8 | 77.3 | 31.0 | 17.0 | 41.7 | 68.4 | 77.9 | 31.0 |
| Ours | 0.90 | 22.1 | 37.8 | 77.3 | 78.1 | 33.4 | 19.6 | 44.8 | 71.9 | 77.7 | 32.4 |
| Pruner | 0.85 | 13.1 | 24.8 | 67.9 | 77.5 | 31.0 | 11.8 | 35.3 | 56.6 | 77.4 | 28.8 |
| Ours | 0.85 | 18.5 | 33.2 | 74.8 | 77.2 | 32.4 | 14.0 | 29.4 | 61.4 | 75.5 | 29.6 |
| Pruner | 0.80 | 7.8 | 25.1 | 64.6 | 76.2 | 29.0 | 5.5 | 23.0 | 53.5 | 74.2 | 24.4 |
| Ours | 0.80 | 13.5 | 29.3 | 71.4 | 75.1 | 32.8 | 8.6 | 24.8 | 53.5 | 74.1 | 25.4 |
| LLM Pruner performance with additional fine-tuning on Alpaca dataset | | | | | | | | | | | |
| Pruner+Finetune | 0.90 | 17.9 | 34.0 | 71.4 | 78.1 | 33.0 | 19.0 | 48.0 | 76.1 | 79.5 | 35.0 |
| Pruner+Finetune | 0.85 | 14.7 | 31.4 | 72.2 | 78.7 | 33.0 | 14.7 | 42.9 | 74.4 | 79.0 | 30.8 |
| Pruner+Finetune | 0.80 | 11.1 | 26.7 | 67.8 | 77.8 | 30.4 | 10.6 | 32.9 | 70.2 | 78.2 | 29.8 |

Table 2: Performance comparison between LLM-Pruner and Our approach on LLAMA-2-7B and LLAMA-3-8B

Pruner without fine-tuning but also outperforms LLM-Pruner with fine-tuning on 4 out of 5 datasets. This demonstrates competitive compression performance on LLaMA-2-7B, even when compared to LLM-Pruner, which uses post-compression fine-tuning. However, for Llama-3-8B, as the compression rate increases, LLM-Pruner with fine-tuning performs stronger than our fine-tuning-free approach. While ARS explores fine-tuning after compression through rank selection, it utilizes much more computation, requiring 576 GPU hours Gao et al. (2024). In our work, we address competitive compression performance in a fine-tuning-free manner and leave approaches integrating further fine-tuning of low-rank compressed models as future work.

# 7 ABLATIONS

This section contains ablations studies that guided modelling design choices.

## 7.1 SELECTING ANY-K SINGULAR VALUES VS TOP-K

Theoretically, when selecting $k$ singular values for retention, the optimal choice to minimize reconstruction loss would typically be the top $k$ singular values by magnitude. However, our approach, which allows the mask to learn to select any $k$ singular values rather than strictly the top $k$, has shown strong practical performance despite limited theoretical backing.

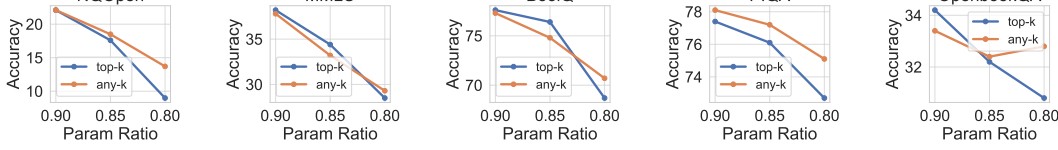

Figure 3: Evaluation performance of Llama-2-7b with using any-k and top-k masking

To evaluate our rank selection method, we compressed the LLaMA-2-7b model to various target parameter ratios and performed evaluation using both the top-$k$ singular value selection and our default any-$k$ method. The results shown in Figure 3 demonstrate, that in the lowest parameter ratio of 0.80, any-$k$ mask outperforms the top-$k$ mask on 5 out of 5 datasets. There are significant gains in open-domain QA with any-k performing  4% better on NQ-Open at parameter ratio 0.80

## 7.2 INTRODUCTION OF TOTAL VARIATION LOSS

Motivated by the theoretical understanding that higher singular values are more relevant, we used the Total Variation (TV) loss to guide our learnt masks to be smooth.

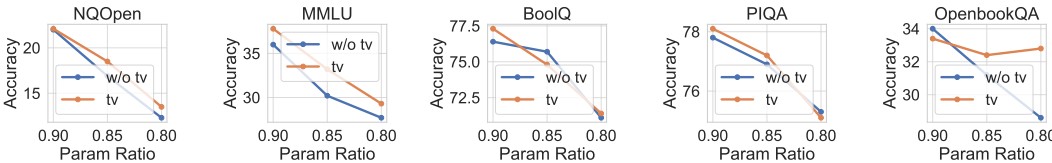

Figure 4: Evaluation performance of Llama-2-7b with and without using total variational loss

To validate this hypothesis, we compressed the LLaMA-2-7b model to various target parameter ratios using and without using the total variational loss. Experimental results shown in Figure 4 demonstrate the efficacy of introducing the total variational (tv) loss function. On NQ-Open, MMLU and PIQA introducing this loss function consistently leads to higher performance. On OpenbookQA, this loss function leads to higher performance on parameter ratios 0.85 and 0.80.

### 7.3 DISTILLATION OBJECTIVE OVER PRE-TRAINING OBJECTIVE

In our final results, we utilized a distillation objective that minimized the divergence between the activations of the compressed model and the original model. While both methods lead to similar results, as shown in Fig. 5, indicating the flexibility of our masking training procedure, learning the optimal ranks through distillation leads to better generalization on the majority of datasets. At 20% compression, distillation outperforms pre-training in all datasets.

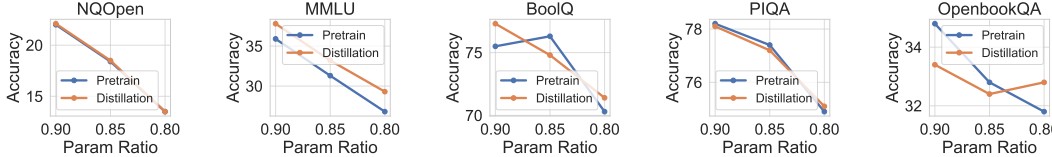

Figure 5: Comparison of performance of models trained using distillation objective and a pre-training objective (next word prediction)

For the pre-training objective, the same dataset was used and documents were packed to a sequence length of 256 tokens. More details of hyper-parameters are in Appendix G.

## 8 CONCLUSION

In summary, our study introduces Learning to Low-Rank Compress (LLRC), an approach that can learn optimal Singular Value Decomposition ranks for each layer from data to perform low-rank compression of language models. The rank selection layer is a learnable parameter vector, making the training computationally more efficient than the related gradient-based rank selection approach called ARS that uses RNNs and linear up-projection layers(Gao et al., 2024). Unique from other rank selection approaches such as Yuan et al. (2024); Gao et al. (2024), which select optimal ranks (*top-k* singular values), our method selects *any-k* singular values to achieve compression. Among rank selection methods, we showed our method applied on Llama-2-7b, Llama-3-8b, Llama-2-13b, and Gemma-7b significantly outperforms all other rank selection approaches on NQ-Open, PIQA, and OpenbookQA. For instance, for Llama-3-8b our method achieved over 6% higher accuracy than the second highest on NQ-Open, PIQA, and OpenBookQA at a parameter ratio of 0.80. On Llama-2-13B, our approach outperforms the competitive STRS on MMLU, BoolQ, and OpenbookQA by 12%, 3.5%, and 4.4%, respectively. Compared to existing compression techniques, our fine-tuning free method outperforms LLM Pruner (without fine-tuning) on various compression rates. More-over, our fine-tuning-free approach also is competitive with LLM-Pruner (with fine-tuning), out-performing it on NQ-Open, MMLU, BoolQ, and OpenbookQA at compression rate of 20% with Llama-2-7B. However, on higher compression rates of 15% and 20% on Llama-3-8B, LLM Pruner (with fine-tuning) performs stronger, indicating the necessity of exploring efficient fine-tuning on low-rank compressed models.

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

# A    TRAINING DETAILS

## A.1    DISTILLATION SCALE PARAMETER

The distillation scale parameter $\alpha$ in Equation 5 equations below:

$$z = \cos\left(\frac{2\pi \cdot 10 \cdot \text{current\_step}}{\text{total\_steps}}\right) \tag{9}$$

$$\alpha = \min\left(\max(z, b), c\right) \tag{10}$$

This scaling $\alpha$ follows a cosine distribution that completes 1 cycle at 10% of total training steps, and the min value is capped at b and max c, which are decided based on the rate of compression required. The lower the value of b and c, the faster the model learns the required compression rate. For llama-2-7b, we use b=0.3 and c=1.0, for llama-3-8b we use b=0.25 and c=0.5, for Gemma-7b we use b=0.5 and c=1.0. As a warmup, for the first 250 steps, we avoid any scaling and use $\alpha = 1.0$.

# B    EVALUATION DETAILS

## B.1    ASVD STRS HYPER-PARAMETERS

The hyperparameters for STRS ASVD are below:

- $\alpha$: 0.5
- Number of Calibration Samples: 32
- Max Sequence Length: 2048

## B.2    ADAPTIVE RANK SELECTION (ARS) HYPER-PARAMETERS

The benchmarks of our ARS algorithm is based on our implementation, which is open-sourced here: GitHub. The dataset used is the same as one used in our work, Wikitext-2 (Merity et al., 2016)

- **Llama-2-7b:** $\lambda$:16, $\gamma$: 1, $\alpha$: 1e-3, Optimizer: Adam
- **Llama-3-8b:** $\lambda$: 8, $\gamma$: 2, $\alpha$: 1e-3, Optimizer: Adam
- **Gemma-7b:**, $\lambda$: 8, $\gamma$: 2, $\alpha$: 1e-3, Optimizer: Adam
- **Llama-2-13b:** $\lambda$:16, $\gamma$: 1, $\alpha$: 1e-3, Optimizer: Adam

For Llama-2-7b, we used $\lambda$=16 and $\gamma$=1, instead of $\lambda$=8 and $\gamma$=2, because the former lead to an acceptable NQ-Open performance at Param Ratio 0.90.

## B.3    LLM PRUNER HYPER-PARAMETERS

This section outlines the hyper-parameters used for benchmarking LLM-Pruner on downstream datasets with various compression ratios.

- **Pruning:**
    - Block-wise Pruning:
        * Block MLP Layer Start: 4
        * Block MLP Layer End: 30
        * Block Attention Layer Start: 4
        * Block Attention Layer End: 30
    - Pruner Type: Taylor
    - Taylor Strategy: `param_first`
- **Post-Training:**
    - Dataset: `yahma/alpaca-cleaned`

- LoRA Rank: 8
- Number of Epochs: 2
- Learning Rate: 1e-4
- Batch Size: 64

## C    INSIGHTS FROM LEARNT COMPRESSION RATES

To gain insights from the learnt compression rates, we exported the learnt singular value masks for all the layers and visualized them in Figure 6. Lower parameter ratios could indicate that the reconstruction of the weight is allowed to be more lossy or has the presence of a strong low-rank structure.

Figure 6 reveals a pattern across all layer types: the earliest layers exhibit a higher degree of compressibility. For example, the up-projection layer in the first layer is compressed by approximately 50%, while later layers show negligible or no compression.

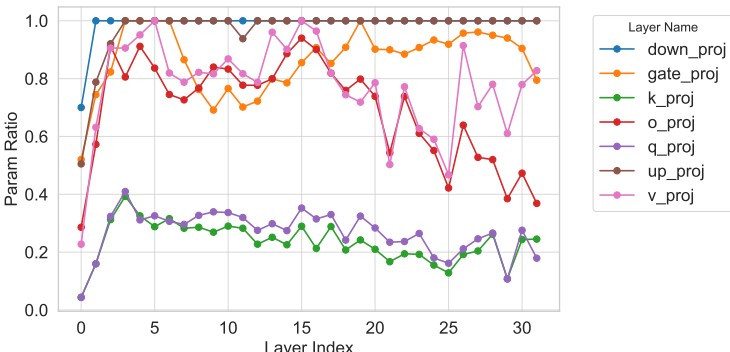

Figure 6: Distribution of compression rates across layer types and layer numbers in the LLaMA-2-7b model that was compressed by 20% (param ratio of 0.80) using our approach.

This finding highlights the distinct nature of the information processed by earlier layers, suggesting a more pronounced low-rank structure in these layers. In addition to the layer-wise compression pattern, the visualization indicates that among the different types of layers, key and query projection layers are the most compressed. This is justifiable, the key and query projection layer only affect the information flow through attention weight scalars rather than through projections. In contrast, up-projection and down-projection layers remain largely uncompressed, suggesting that the model suffers more by compressing these layers. This distinction further underscores the variability in the representational capacity and functional roles of different layers within the model.

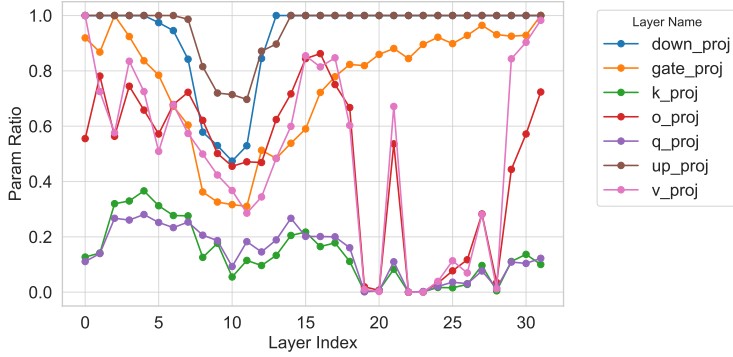

Figure 7: Distribution of compression rates across layer types and layer numbers in the LLaMA-3-8b model that was compressed by 20%

The learn compression rates in Llama-3-8B, as shown in Figure 7, differ from Llama-2-7B's, in that the key and query project layers are compressed much more. One explanation for this is the difference in architecture. The architecture of Llama-3-8B contrasts with Llama-2-7B in that there is a greater number of parameters in the feedforward network. In llama-3-8B, the up projection layer projects from a dimension of 4096 to 14,336, whereas llama-2-7B projects 4096 to 11,008. Figure 6 showed us that the up_projection and down_projection layers are the least compressible in Llama-2-7B. Hence, with the increased size of the up_projection and down_projection layers in Llama-3-8B, the training procedure is forced to compress keys and values even more.

## D   PERFORMANCE ON GEMMA-7B

This section contains the table of metrics of Gemma-7B.

| Method | Param Ratio | NQ-Open | MMLU | BoolQ | PIQA | OpenbookQA |
|--------|-------------|---------|------|-------|------|------------|
| Original | 1.00 | 25.7 | 62.0 | 83.4 | 80.1 | 32.4 |
| Fixed Rate | 0.90 | 2.88 | 33.8 | 71.3 | 67.8 | 26.6 |
| STRS | 0.90 | 14.3 | 45.4 | 81.5 | 78.2 | 31.4 |
| ARS | 0.90 | 15.3 | 29.4 | 67.2 | 76.7 | 30.2 |
| **Ours** | 0.90 | 13.2 | 49.6 | 77.2 | 76.1 | 32.2 |
| Fixed Rate | 0.85 | 0.39 | 27.4 | 62.6 | 59.4 | 19.4 |
| STRS | 0.85 | 2.30 | 29.3 | 70.0 | 64.4 | 27.6 |
| ARS | 0.85 | 7.40 | 24.3 | 60.7 | 73.7 | 27.4 |
| **Ours** | 0.85 | 11.3 | 37.0 | 78.3 | 75.1 | 34.4 |
| Fixed Rate | 0.80 | 0.14 | 23.0 | 56.8 | 57.0 | 18.8 |
| STRS | 0.80 | 0.11 | 23.4 | 51.2 | 60.0 | 21.6 |
| ARS | 0.80 | 2.80 | 23.1 | 54.0 | 71.3 | 27.2 |
| **Ours** | 0.80 | 8.39 | 33.6 | 75.8 | 74.7 | 34.6 |

Table 3: Evaluation performance of Gemma-8B using different rank selection methods

## E   PERFORMANCE ON LLAMA-2-13B

| Method | Param Ratio | NQOpen | MMLU | BoolQ | PIQA | OpenbookQA |
|--------|-------------|--------|------|-------|------|------------|
| Original | 1.00 | 30.7 | 52.1 | 80.1 | 79.1 | 35.2 |
| Fixed | 0.90 | 13.2 | 41.2 | 79.6 | 75.0 | 32.0 |
| STRS | 0.90 | 27.6 | 49.1 | 80.6 | 78.0 | 35.8 |
| ARS | 0.90 | 11.1 | 39.9 | 74.0 | 75.8 | 30.2 |
| **Ours** | 0.90 | 26.5 | 49.7 | 81.1 | 78.6 | 35.4 |
| Fixed | 0.85 | 10.1 | 36.1 | 77.9 | 73.6 | 29.6 |
| STRS | 0.85 | 23.7 | 44.9 | 78.8 | 78.3 | 35.2 |
| ARS | 0.85 | 7.22 | 38.3 | 71.3 | 72.0 | 23.8 |
| **Ours** | 0.85 | 22.7 | 48.1 | 80.5 | 77.4 | 36.0 |
| Fixed | 0.80 | 6.4 | 36.3 | 77.7 | 71.5 | 27.8 |
| STRS | 0.80 | 17.5 | 32.1 | 77.8 | 75.6 | 31.6 |
| ARS | 0.80 | 3.13 | 24.7 | 63.2 | 67.6 | 23.8 |
| **Ours** | 0.80 | 18.9 | 44.1 | 81.3 | 77.9 | 36.0 |

Table 4: Evaluation performance of Llama-2-13B using different rank selection methods

## F    IMPLEMENTATION CAVEATS OF ARS

Our benchmarking process of ARS Gao et al. (2024) involved a slight variation from the original method, where we included the entire hypernetwork, including the GRU network, for every layer required to be compressed. Based on feedback, this approach was later adjusted to align with the original paper, where only a single GRU hypernetwork was used for the entire model but separate linear projection layers for every layer required to be compressed. Despite this adjustment, results did not change significantly to affect the competitiveness of the approach.

## G    PRE-TRAINING OBJECTIVE HYPER-PARAMETERS

The hyperparameters used in the training process for the pre-training objective are summarized in this section. We sampled 70,000 documents from the Wikitext dataset (Merity et al., 2016), which was the same data source used for the model trained on the distillation objective. We utilized document packing, ensuring that all documents contained 256 tokens. Besides this, all parameters, such as optimizer, maximum number of tokens, learning rate, and early stopping criteria, were the same as the experiment that used distillation, as mentioned in Section 5.2.

