# OpenReview forum: "Low-Rank Compression of Language Models Via Differentiable Rank Selection"
_ICLR.cc/2025/Conference — Submitted to ICLR 2025_

### Official Review · Reviewer_RHXN · 2024-10-16

**Soundness:** 2
**Presentation:** 1
**Contribution:** 1
**Rating:** 3
**Confidence:** 4

**Summary:**

This paper focuses on the rank selection problem in the context of low-rank decomposition in language model compression. Prior researches on low-rank decomposition mainly focus on better reconstructing the weight matrix or output activation while assuming the same compression rate is shared across all modules. Instead, this work proposes LLRC(Learning to Low-Rank Compress), which insert learnable masks into each linear layer for selecting singular values for compression. The method starts with Activation-aware SVD to obtain the initial factorized form of weight matrices, then utilize Gumbel Sigmoid to transform mask variables into continuous binary mask. The training objective consists of three sub-parts, which are Compression Loss, Distillation Loss, And Total Variation Loss. After training on a 3,000 documents calibration dataset, the learned masks variables are used for selecting the ultimate singular values to preserve. Experiment evaluation are performed using Base LLM(LLaMa2-7B and LLaMa3-8B) on five zero-shot commonsense reasoning and question answering tasks. Results demonstrate moderate improvements compared to previous rank selection methods.

**Strengths:**

1. The training process of LLRC is efficient, as it only requires gradient updates on the singular value masking variables.
2. LLRC is able to adaptively allocate rank budget to weight matrices, allowing for flexible model parametrization.

**Weaknesses:**

1. The writing of this paper needs significant improvement. There are considerable amount of typos, grammar error, and formatting issues in the submission, e.g., Line 94, Line 195, Line 226, Line 262, Line 355, Line 452, Figure 2&3(out of margin). Also note the repetitive description as in Line 255-257 and Line 284-286.
2. The novelty of LLRC is limited. The idea of using learnable pruning mask variables to select ranks of SVD-based factorization has been previously explored[1]. The major difference only lies in LLRC is applied on LLM and on task-agnostic setting.
3. The downstream performance of compressed model is unsatisfactory, given that the compression rate is merely 20%. Moreover, LLRC does not exhibit consistent superiority over prior rank selection methods, according Table 1.
4. This paper lacks  experimental validation of the effectiveness of LLRC on instruction-tuned models besides the base versions.
5. The compression rate reported in the submission is relatively low and the performance degradation induced by compression clearly outweigh the efficiency gain.



[1]. Structured Pruning of Large Language Models. *EMNLP 2020*.

**Questions:**

See weakness above.

---

> ### Author Response · Authors · 2024-11-20
> **Response**
>
> Thank you for your helpful review. Please find our responses below:
>
> * Weakness #1: The writing of this paper needs significant improvement. There is a considerable amount of typos, grammar errors, and formatting issues in the submission, e.g., Line 94, Line 195, Line 226, Line 262, Line 355, Line 452, Figure 2&3(out of margin). Also note the repetitive description as in Line 255-257 and Line 284-286.
>
> An earlier version of the paper was inadvertently uploaded at the time of submission, which contained several typos and formatting issues. We apologize for this mistake. In the updated version we have submitted, we addressed all these errors, including typographical corrections, formatting adjustments, and removal of repetitive descriptions. Thank you for pointing these out.
>
>
> * Weakness #2: The novelty of LLRC is limited. The idea of using learnable pruning mask variables to select ranks of SVD-based factorization has been previously explored [1]. The major difference only lies in LLRC is applied on LLM and on task-agnostic setting.
>
> We acknowledge that the concept of using learnable pruning masks for SVD-based factorization has been explored in prior work such as [1]. We have now referenced it in our work. However, it explores it in conjunction with expensive training of the entire network and applies it to smaller language models. Our approach introduces several distinctive novelties and advancements: it is specifically designed for a fine-tuning-free setting with LLMs, incorporates weighted singular value decomposition, and employs novel techniques for refining learned ranks, including top-k/any-k masking and targeted distillation.
>
> [1] Structured Pruning of Large Language Models. EMNLP 2020.
>
> * Weakness #3: The downstream performance of the compressed model is unsatisfactory, given that the compression rate is merely 20%. Moreover, LLRC does not exhibit consistent superiority over prior rank selection methods, according to Table 1.
>
> To further assess the effectiveness of our proposed compression method, we extended our evaluation to two additional models: Llama-2-13B and Gemma-7B; our experiments are summarised in Figure 2. For both models, at our highest compression rate (20%), our approach consistently yields more accurate results than STRS across all datasets. For Llama-2-13B at 20% compression, our approach achieves significant improvements over STRS, with gains of 12%, 3.5%, and 4.4% on the MMLU, BoolQ, and OpenBookQA datasets, respectively. A general trend is that our method leads to more stable and reliable performance across various datasets. While our approach significantly outperforms STRS on several datasets, there are no instances where STRS yields more accurate results than our method by such large margins. This suggests that our method yields more accurate and robust results. This pattern can be clearly seen when looking at plots for Llama-3-8B and Llama-2-13B in Figure 2.
>
>
> * Weakness #4: The compression rate reported in the submission is relatively low and the performance degradation induced by compression clearly outweigh the efficiency gain.
>
> In this work, we focus on improving the downstream task accuracy of compressed low-rank decomposed large language models (LLMs) via learnable rank selection without requiring data-intensive fine-tuning procedures. We demonstrate that our approach yields more accurate results than existing rank selection methods across a variety of models. In contrast, methods like Adaptive Rank Selection (ARS) [1] or Sheared LLaMA [2], while achieving competitive performance, rely on a second stage of expensive post-compression training to recover performance. After compression in the first stage, ARS performs continued pre-training for 576 GPU/hours [1] while Sheared LLaMA performs pre-training on 50B tokens [2]. To this end, our goal was to improve the performance of the compressed model without requiring data-intensive fine-tuning steps in the first stage. For the second stage, we leave the exploration of efficient methods to recover performance in low-rank decomposed models to improve the compression-performance tradeoff to future work.
>
> [1] Adaptive Rank Selections for Low-Rank Approximation of Language Models, (https://aclanthology.org/2024.naacl-long.13.pdf)
> [2] Sheared LLaMA: Accelerating Language Model Pre-training via Structured Pruning (https://arxiv.org/abs/2310.06694)
>
>
> * Weakness #5: This paper lacks experimental validation of the effectiveness of LLRC on instruction-tuned models besides the base versions:
>
> This is a good point. We focused on four base models here, arguing that gains will transfer to fine-tuned models. We can aim to add additional experiments on instruction-tuned models in the paper

---

> > ### Comment · Reviewer_RHXN · 2024-11-21
> >
> > Thanks for your responses. I appreciate that the authors updated their manuscript to fix typo and formatting issues. My remaining concerns after reading the authors' response is listed below:
> >
> > Regarding novelty: I acknowledge that the proposed LLRC is distinct from prior work that adopt learnable pruning mask variables to select ranks of SVD-based factorization. However, from my understanding, the differences mainly manifest as (1) freeze other parts of LLMs while only tuning rank selection variables; (2) incorporating distillation and total variation loss. For the first point, it cannot be considered as novelty, but instead a different application scenario. For the second point, adding distillation objective has been extensively explored in prior model pruning(either structured or unstructured) literatures.
> >
> > Regarding LLMs selected for compression: Generally, model compression is more effective on larger scale LLMs because the higher redundancy within their parameters. For 7/8B scale models like LLaMa2-7B, Gemma-7B, LLaMa3-8B, the accuracy drop is truly non-negligible for practical usage(T). Therefore, if resource permits, I suggest the authors conduct experiments mainly on larger-scale LLMs to examine if LLRC can reach a higher compression ratio without significant performance drop.
> >
> > Regarding efficiency gains: Can the authors provide statistics about the efficiency gain after applying compression?

---

> ### Author Response · Authors · 2024-11-21
>
> Thanks for your review again.
>
> **Regarding LLMs selected for compression:**
> Regarding your first point about non-negligible performance drop on 7B/8B parameter models: We understand this concern and offer an alternate viewpoint- we see this as a common effect seen across even other compression methods. Other works such as ARS and Sheared Llama offset poor performance in stage 1 of pruning by expensive pre-training in stage 2. We aimed to improve stage 1 performance to offset some load required in stage 2.
>
> Regarding your point about testing on larger models: To test compression performance on larger sizes, over 8B, we have evaluated the compression performance (of 4 rank selection methods) on a larger 13 billion parameter model: *LLama-2-13B*. As you pointed out, larger models have higher redundancy within their parameters and we find that the performance drop due to compression can be lesser- for eg: for 15% compression, the *percent change* w.r.t its original model in MMLU is -7.7% with Llama-2-13B and -28% with Llama-2-7B.
>
> To further understand the behaviour of low-rank compression on larger models, over 8B, we can aim to test on more to complement our existing results on Llama-2-13B.
>
> **Regarding efficiency gains:**
> We will double check and get back to you about this. We did consider understanding efficiency impacts and ran a few tests earlier on, and remember noticing no gains in latency (20% compressed Llama-2-7B). That being said, we will need to double-check and get back to you on this.

---

### Official Review · Reviewer_1JwN · 2024-11-02

**Soundness:** 2
**Presentation:** 2
**Contribution:** 2
**Rating:** 5
**Confidence:** 3

**Summary:**

The authors present LLRC, a novel approach for compressing large language models (LLMs) using adaptive low-rank decomposition via Singular Value Decomposition (SVD). LLRC introduces a differentiable rank selection mechanism that dynamically identifies the optimal compression rate per layer, balancing performance retention and model efficiency.

**Strengths:**

1. The use of multi-objective loss functions, including distillation and total variation loss, helps LLRC retain high performance even at high compression rates, making it effective for deployment in resource-constrained environments.

2. By freezing the main model weights and only learning the mask layer, LLRC reduces the computational burden during training, making it more efficient than traditional compression methods.

**Weaknesses:**

1. The method is similar to structured pruning approaches, such as Sheared LLaMA (https://arxiv.org/abs/2310.06694), yet these works are neither cited nor compared against, which limits the paper’s contextual grounding.

2. When comparing with pruning and distillation methods, the paper does not choose the most competitive or state-of-the-art approaches, making it unclear how LLRC performs against the best available compression techniques.

3.Based on my own empirical experience, datasets like BoolQ and PIQA often exhibit high variance in performance, which can obscure the true effectiveness of a compression method. In contrast, MMLU is generally more scientifically consistent and reliable for evaluating language models. The paper shows that the proposed method does not yield a notable improvement over STRS on MMLU, and in fact, it lags behind STRS on the LLaMA-3-8B model. This limitation on a stable and rigorous benchmark like MMLU raises questions about the robustness of the method, particularly when applied to more demanding or scientifically rigorous evaluation tasks.

**Questions:**

1. Given that datasets like BoolQ and PIQA are known for high variance in performance across different checkpoints, how did you approach checkpoint selection for these evaluations? Did you adopt any specific strategy, such as averaging performance across multiple checkpoints or selecting based on a validation set, to mitigate potential fluctuations and ensure fair comparison across methods?

---

> ### Author Response · Authors · 2024-11-20
> **Response**
>
> Thank you for your review. Please find our responses to your points below:
>
> * Weaknesses #1: The method is similar to structured pruning approaches, such as Sheared LLaMA (https://arxiv.org/abs/2310.06694), yet these works are neither cited nor compared against, which limits the paper’s contextual grounding.
>
> Thank you for bringing this to our attention. Sheared LLaMA employs a structured pruning process followed by a computationally intensive fine-tuning stage, utilizing 0.4 billion tokens during pruning and 50 billion tokens for further pre-training [1]. In comparison, LLM-Pruner adopts an efficient pruning step followed by a parameter-efficient fine-tuning (with LoRA) on a dataset of 50k documents, which requires just three hours [2]. As our method is entirely fine-tuning-free, we focused on comparisons with lightweight compression techniques, in particular those that also forgo fine-tuning. However, we agree that Sheared LLaMA is a relevant work in this space and have added it to the related work section to provide further context.
>
> [1] Sheared LLaMA: Accelerating Language Model Pre-training via Structured Pruning (https://arxiv.org/abs/2310.06694)
> [2] LLM-Pruner: On the Structural Pruning of Large Language Models (https://arxiv.org/pdf/2305.11627)
>
> * Weakness #2: When comparing with pruning and distillation methods, the paper does not choose the most competitive or state-of-the-art approaches, making it unclear how LLRC performs against the best available compression techniques.
>
> Thank you for this feedback. One of our main objectives was to improve the downstream accuracy of low-rank decomposed models by improving rank selection while specifically maintaining the advantage of fine-tuning-free approaches. Consequently, we focused our comparisons on fine-tuning-free compression methods, which narrowed the scope of methods selected. For structured, fine-tuning-free approaches, our comparisons include state-of-the-art techniques. We also provide LLM-Pruner as an additional valuable baseline, allowing readers to compare with both fine-tuning-free and fine-tuning-based variants with relatively low computational costs.
>
> If your comment refers to quantisation-based methods, we would like to clarify that, consistent with LLM-Pruner’s approach, we view quantisation (which indeed achieves stronger compression/performance rates) as a technique that can be used jointly with layer factorization rather than one to be directly compared against.
>
> * Weakness #3: Based on my own empirical experience, datasets like BoolQ and PIQA often exhibit high variance in performance, which can obscure the true effectiveness of a compression method. In contrast, MMLU is generally more scientifically consistent and reliable for evaluating language models. The paper shows that the proposed method does not yield a notable improvement over STRS on MMLU, and in fact, it lags behind STRS on the LLaMA-3-8B model. [Remaining part of review]
>
> For MMLU, as shown in Figure 2, our method demonstrates competitive performance in comparison with STRS on LLaMA-2-7B, LLaMA-3-8B and LLaMA-2-13B. Notably on LLaMA-2-13B, which we added in this new submission to benchmark more models, our approach significantly yields more accurate results than STRS on MMLU, highlighting the robustness of our method on larger models.  For instance, on Llama-2-13B, Figure 2 and Table 4 show that our approach leads to 12% higher accuracy on MMLU compared to STRS at a compression rate of 20%
>
> Additionally, in text generation and factual knowledge benchmarks like NQOpen, our method consistently outperforms STRS on all models - achieving a notable 4.45% improvement on 20%-compressed LLaMA-2-7B.
> .
> * Question #1: Given that datasets like BoolQ and PIQA are known for high variance in performance across different checkpoints, how did you approach checkpoint selection for these evaluations? Did you adopt any specific strategy, such as averaging performance across multiple checkpoints or selecting based on a validation set, to mitigate potential fluctuations and ensure fair comparison across methods?
>
> Thank you for raising this interesting point. Since our proposed method is fine-tuning-free, it does not require checkpoint selection. After training, the learnt masks are used to reduce the layer ranks of the model. Similarly, other baselines like STRS and fixed rate also do not require checkpoint selection- when these approaches are applied, ranks are selected per layer. After that, compression is performed using only those ranks. We also refer to relevant research from [1] and [2], which also evaluate BoolQ and PIQA but do not mention any checkpoint selection process.
>
> [1] Adaptive Rank Selections for Low-Rank Approximation of Language Models, (https://aclanthology.org/2024.naacl-long.13.pdf)
> [2] LLM-Pruner: On the Structural Pruning of Large Language Models (https://arxiv.org/pdf/2305.11627)

---

> > ### Comment · Reviewer_1JwN · 2024-11-25
> >
> > Thanks for your responses.  I decide to maintain my score.

---

> ### Author Response · Authors · 2024-11-24
> **Follow-up: Any feedback on Rebuttal?**
>
> Given the deadline for discussions on the rebuttal is the 26th, we wanted to re-check if you had any thoughts on our response above?
>
> We've submitted a new version of the submission, using your valuable feedback and shared the points above addressing the concerns

---

### Official Review · Reviewer_PNt2 · 2024-11-03

**Soundness:** 2
**Presentation:** 3
**Contribution:** 2
**Rating:** 5
**Confidence:** 4

**Summary:**

This paper introduces a novel approach to compressing large language models (LLMs) using low-rank approximation. The key innovation is the introduction of a learnable masking mechanism within the Singular Value Decomposition (SVD), which dynamically selects eigenvalues and eigenvectors during the low-rank approximation process. Unlike traditional methods that rely on a fixed rank and select only the top-K components with the largest eigenvalues, this approach uses learnable masking parameters to optimize rank allocation based on loss constraints. This flexibility allows the model to allocate rank to components with smaller eigenvalues when beneficial, making the compression more adaptive and potentially more effective.

**Strengths:**

1. Clear idea. The motivation is good and the solution is reasonable.
2. The writing is clear and easy to follow

**Weaknesses:**

My primary concern is the trade-off between compression and model quality. While the model achieves some compression, it sacrifices quality significantly. A 20% reduction in parameters leads to a notable degradation in performance. Although the paper attempts to show improvements over certain baselines, a more meaningful comparison would be with a non-compressed model of similar size. For instance, if the method can compress an 8B model to 3B while still outperforming a standard 3B model, it would be valuable in practice. Otherwise, the utility of the compressed model for real-world applications seems limited. I recommend the authors address this comparison in Table 1 and incorporate a discussion on it within the paper.

Additionally, among the various benchmarks, MMLU deserves particular attention, as its results are mixed when comparing the proposed approach with STRS. The substantial drop in MMLU performance, despite the limited compression rate, raises concerns about the effectiveness of the method.

Certain design choices in the paper also need clarification. For example, the paper applies an average of the learnable weights in the L_compression loss, yet averaging can be highly sensitive to extreme negative values. Exploring alternative approaches could enhance robustness, and it would strengthen the paper if the authors could justify why this choice is optimal.

**Questions:**

see weakness

---

> ### Author Response · Authors · 2024-11-20
> **Response to feedback**
>
> Thank you for your helpful review! Please find our responses to your review below
>
> * Weaknesses #1: My primary concern is the trade-off between compression and model quality.Although the paper attempts to show improvements over certain baselines, a more meaningful comparison would be with a non-compressed model of similar size. [Rest]
>
> In this work, we focus on improving the downstream task accuracy of compressed low-rank decomposed large language models (LLMs) via learnable rank selection without requiring data-intensive fine-tuning procedures. We demonstrate that our approach yields more accurate results than existing rank selection methods across a variety of models. In contrast, methods like Adaptive Rank Selection (ARS) or Sheared LLaMA, while achieving competitive performance and greater compression, rely on a second stage of expensive post-compression training to recover performance. After compression in the first stage, ARS performs continued pre-training for 576 GPU/hours [1] while Sheared LLaMA performs pre-training on 50B tokens [2]. To this end, our goal was to improve the performance of the compressed model without requiring data-intensive fine-tuning steps in the first stage. For the second stage, we leave the exploration of efficient methods to recover performance in low-rank decomposed models to improve the compression-performance tradeoff to future work.
>
> [1] Adaptive Rank Selections for Low-Rank Approximation of Language Models, (https://aclanthology.org/2024.naacl-long.13.pdf)
> [2] Sheared LLaMA: Accelerating Language Model Pre-training via Structured Pruning, (https://arxiv.org/abs/2310.06694)
>
>
>
> * Weaknesses #2: Additionally, among the various benchmarks, MMLU deserves particular attention, as its results are mixed when comparing the proposed approach with STRS. The substantial drop in MMLU performance, despite the limited compression rate, raises concerns about the effectiveness of the method.
>
> To further evaluate the effectiveness of our proposed approach, we extended our benchmarking to two additional models: Llama-2-13B and Gemma-7B. Our method outperforms STRS on MMLU across various compression rates with both models. For instance, on Llama-2-13B, Figure 2 and Table 4 show that our approach leads to 12% higher accuracy on MMLU compared to STRS at a compression rate of 20%.
>
> Moreover, we observed that the performance drop on MMLU due to compression differs across models. For instance, compressing Llama-2-13B by 20% results in only an 8% performance reduction on MMLU, whereas the same compression rate in Llama-2-7B leads to a 12% drop. While our approach was designed to improve compression performance without data-intensive fine-tuning, we hypothesise that this trade-off on MMLU could potentially be placated through additional fine-tuning efforts.
>
> * Weaknesses #3: Certain design choices in the paper also need clarification. For example, the paper applies an average of the learnable weights in the L_compression loss, yet averaging can be highly sensitive to extreme negative values. Exploring alternative approaches could enhance robustness, and it would strengthen the paper if the authors could justify why this choice is optimal.
>
> In our preliminary analyses, we also experimented with different choices of compression losses. We found the one proposed in Eq. 6 to yield accurate results while also being robust to different values of $\beta$. We will expand on this preliminary analysis in the camera-ready version.

---

> ### Author Response · Authors · 2024-11-24
> **Follow-up: Any feedback on Rebuttal?**
>
> Given the deadline for discussions on the rebuttal is the 26th, we wanted to re-check if you had any thoughts on our response above?
>
> We've submitted a new version of the submission, using your valuable feedback and shared the points above addressing the concerns

---

### Official Review · Reviewer_xjEE · 2024-11-04

**Soundness:** 3
**Presentation:** 2
**Contribution:** 2
**Rating:** 5
**Confidence:** 3

**Summary:**

This paper aims to improve the performance of low-rank compression for LLM. To achieve this goal, they propose learning optimal ranks on 3,000 articles using multiple loss functions. Experimental results show that it performs better than previous SVD methods.

**Strengths:**

- The learnable mask mechanism has a higher potential compared to heuristic algorithms.

- The experimental results are positive, outperforming previous SVD methods

**Weaknesses:**

- sloppy formatting. The reference to the table is missing in line 452. AUTHOR CONTRIBUTIONS and ACKNOWLEDGMENTS sections have not been removed. The authors should check their manuscript more carefully.
- cannot outperform LLM Pruner. LLM Pruner requires only a small amount of data (e.g., 128 articles) for fine-tuning to achieve better results.
- no ablation study on multiple loss functions. The authors used three loss functions but did not verify their effects in the experiments.

**Questions:**

What is the setting of LLM Pruner in Table 2? LLM Pruner has multiple configurations, and the authors did not specify which one was used. Additionally, if training is performed after compression (like LLM Pruner), can better results be obtained?

---

> ### Author Response · Authors · 2024-11-20
> **Response to feedback**
>
> We thank the reviewer for the helpful comments and feedback
>
> * Weaknesses # 1: sloppy formatting. The reference to the table is missing in line 452. AUTHOR CONTRIBUTIONS and ACKNOWLEDGMENTS sections have not been removed. The authors should check their manuscript more carefully.
>
> We apologize for the oversight. Unfortunately, we accidentally uploaded an earlier version of the paper. We have now replaced it with an updated version that corrects these issues, including the missing table references and other formatting inconsistencies. Thank you for bringing these to our attention.
>
> * Weaknesses # 2: cannot outperform LLM Pruner. LLM Pruner requires only a small amount of data (e.g., 128 articles) for fine-tuning to achieve better results.
>
> Thank you for your feedback. We identified an error in our initial LLM-Pruner experiments due to an issue in the two-stage command sequence for Pruning and Pruning + Fine-tuning: while adapting the codebase for logging intermediate results, an incorrect model was loaded for the Pruning + Fine-tuning step. After correcting this, we regenerated the results table, Table 2. As previously observed, our fine-tuning-free approach consistently outperforms LLM Pruner without fine-tuning. Notably, after this correction which fixed the fine-tuning step, our method also proves competitive with LLM Pruner + Fine-tuning. Specifically, our fine-tuning-free approach outperforms LLM Pruner + Fine-tuning on 4 out of 5 datasets with LLaMA-2-7B (20% compression) and on 3 out of 5 datasets with LLaMA-2-7B (15% compression).
>
> Additionally, we wish to clarify that the official implementation of LLM Pruner we used, with fine-tuning, uses a dataset of 51,800 documents (from [1]) rather than just 128 documents. This follows the fine-tuning process described in the LLM-Pruner paper [2], where authors fine-tune on ~50k documents.
>
> [1] https://huggingface.co/datasets/yahma/alpaca-cleaned.
>
> [2] Ma, Xinyin et al. LLM-Pruner: On the Structural Pruning of Large Language Models (https://arxiv.org/abs/2305.11627)
>
> * Weaknesses #3: no ablation study on multiple loss functions
>
> Thank you for pointing this out. The ablation study we added in Section 7.2 does demonstrate the effectiveness of including the Total Variation (TV) loss function. We observed that incorporating the TV loss leads to improved and more consistent performance across different compression rates. For example, at a 20% compression rate on OpenbookQA, the model achieves a 2.6% improvement with the addition of TV loss. We also added ablation, in Section 7.3, using a pre-training loss, with next-word prediction, and compared it to distillation. We will clarify these two in the revised version of the paper.
>
>
> * Question #1:  What is the setting of LLM Pruner in Table 2? LLM Pruner has multiple configurations, and the authors did not specify which one was used. Additionally, if training is performed after compression (like LLM Pruner), can better results be obtained?
>
> This setting is present in Appendix B.3. This contains all the hyperparameters and the dataset used in LLM Pruner and LLM Pruner + Finetuning. Yes, for LLM Pruner, as presented in Table 2, we also report results from LLM-Pruner fine-tuning after the compression. Remarkably, we notice that for Llama-2-7B, even after LLM Pruner was applied and fine-tuned, our fine-tuning free approach outperforms it on 4 out of 5 datasets at a compression rate of 20%.

---

> ### Author Response · Authors · 2024-11-24
> **Follow-up: Any feedback on Rebuttal?**
>
> Given the deadline for discussions on the rebuttal is the 26th, we wanted to re-check if you had any thoughts on our response above?
>
> We've submitted a new version of the submission, using your valuable feedback and shared the points above addressing the concerns

---

> > ### Comment · Reviewer_xjEE · 2024-11-24
> > **Response to authors**
> >
> > Thanks for your response. Regarding question#1, I meant whether further fine-tuning your method could lead to better results.

---

> > > ### Author Response · Authors · 2024-11-24
> > > **Response**
> > >
> > > Understood. For this question, we want to understand if further fine-tuning can lead to better results on our low-rank compressed model:
> > >
> > > We would say yes, depending on the method. If we are performing expensive continued fine-tuning on the compressed model, it can improve:
> > > related work such as ARS [1] have demonstrated that continued pretraining (576 GPU hours for Llama-7b) shows strong performance at high compression (Table 6 [1]).
> > >
> > > That being said, these approaches that perform competitive compression such as ARS [1] or Sheared Llama [2], perform stage 1 of lossy compression and stage 2 of expensive fine-tuning. We aimed to improve stage 1. Now, we also want to figure out how to more efficiently recover performance in low-rank decomposed models, rather than using the ARS route of computationally heavy fine-tuning (576 GPU hours).
> > >
> > > [1] Adaptive Rank Selections for Low-Rank Approximation of Language Models, (https://aclanthology.org/2024.naacl-long.13.pdf)
> > > [2] Sheared LLaMA: Accelerating Language Model Pre-training via Structured Pruning, (https://arxiv.org/abs/2310.06694)

---

### Author Response · Authors · 2024-11-21
**Summarised response**

We thank all the reviewers for their helpful feedback. Apart from the individual responses we provided yesterday, we are sharing one summarised response addressing the primary concerns which we addressed in the updated rebuttal paper version:

Overall, we identified three primary concerns raised: 1) writing and formatting, 2) competitiveness with other rank selection techniques, and 3) compression performance.

**Writing/Formatting**

We have addressed the issues related to formatting and writing style to improve clarity and presentation.

**Competitiveness with Other Rank Selection Techniques**

Initially, we benchmarked our rank selection approach on two models. For this rebuttal, we extended our evaluation to two additional models to comprehensively assess its performance. The models now include Llama-2-7B, Llama-3-8B, Llama-2-13B, and Gemma-7B. Based on the results in Figure 2, our method consistently outperforms competing techniques across the majority of datasets. For example, when evaluating Llama-2-13B at 20% compression, our approach demonstrates significant gains over STRS, achieving improvements of 12%, 3.5%, and 4.4% on the MMLU, BoolQ, and OpenBookQA datasets, respectively. Out of the 20 combinations of metrics, resulting from 4 models and 5 datasets at a compression rate of 20%, our method performs the best on 17/20 cases.

Moreover, a general trend we observed in Figure 2 is that our method delivers more stable and reliable performance across diverse datasets. While our approach shows substantial improvements over the competitive STRS on datasets such as MMLU (12% gain on Llama-2-13B at 20% compression), NQ-Open (4.3% gain on Llama-2-7B at 20% compression), and OpenBookQA (8.8% gain on Llama-3-8B at 20% compression), there are no instances where STRS achieves higher accuracy by similarly large margins. Overall, we emphasize the strong and consistent compression performance of our method across various downstream tasks compared to existing approaches.

**Compression Performance**

The second key concern relates to compression performance from two perspectives: comparison with efficient pruning techniques (e.g., LLM Pruner) and general performance. In response to the first, our updated results demonstrate that our fine-tuning-free approach outperforms LLM-Pruner and is competitive even with LLM-Pruner+Finetuning, outperforming it with Llama-2-7B. For instance, on Llama-2-7B at 20% compression, our method outperforms LLM-Pruner+Finetuning on 4 out of 5 datasets.

In response to the second, we acknowledge that compression using all rank selection techniques (fine-tuning-free) inevitably results in more performance degradation compared to fine-tuning-based methods, such as Sheared Llama or ARS, which rely on extensive continued pretraining (e.g., 576 GPU hours for ARS and 50B tokens for Sheared Llama). Despite this, among low-rank compressed LLMs, our rank selection approach consistently achieves superior performance.

Competitive compression methods like ARS typically involve a two-stage pipeline: an initial very lossy compression step followed by an expensive fine-tuning process to recover performance. Our goal was to enhance the first stage of this pipeline by improving the performance of low-rank decomposed models prior to fine-tuning. Moving forward, we also recognise the importance of exploring efficient fine-tuning strategies to further recover performance in the second stage. Therefore, after strengthening stage 1, our future work will focus on developing effective and efficient fine-tuning practices to complement our approach.

---

### Meta-Review · Area_Chair_wPqx · 2024-12-19

**Metareview:**

The paper proposes a method called LLRC that learns the optimal ranks for low-rank decomposition of large language models (LLMs) in a fine-tuning-free manner. LLRC uses learnable masks to select the singular values to retain during the low-rank approximation, which is trained on a small calibration dataset.

*The key strengths*
1. The efficient training process that only requires updating the learnable mask parameters,
2. The adaptive rank allocation mechanism that can flexibly allocate rank budget across different weight matrices.

*Main weaknesses*
1. The writing and formatting needs significant improvement with many typos and formatting issues,
2. The novelty of the approach is limited as the core idea of learnable pruning masks has been explored before,
3. The compression performance and downstream task accuracy are not consistently superior to prior methods, especially on the MMLU benchmark,
4. Lack of experiments on larger LLM architectures beyond 8B parameters, and
5. Lack of analysis on the efficiency gains from the compression.

Overall, while the core idea has some merit, the reviewers do not find the current submission to be strong enough to warrant acceptance, given the issues with writing quality, novelty, and experimental validation.

**Additional Comments On Reviewer Discussion:**

During the rebuttal period, the authors responded to the reviewers' feedback in detail. They acknowledged the issues with the initial submission, such as formatting and writing problems, and confirmed that they had addressed these in the updated version. Regarding the novelty concerns, the authors argued that while the core idea of learnable pruning masks has been explored before, their specific application to LLM compression and the incorporation of techniques like weighted SVD and targeted distillation represent meaningful advancements. To address the performance concerns, the authors expanded their experiments to include larger LLM models like Llama-2-13B and Gemma-7B, demonstrating that their approach consistently outperforms prior rank selection methods like STRS across a variety of tasks and compression rates. They also provided clarification on the efficiency gains and discussed plans to further explore efficient fine-tuning methods to complement their compression approach.

Unfortunately, the reviewers remain unconvinced, resulting in generally negative final ratings.

---

### Decision · Program_Chairs · 2025-01-22

Reject